# Correlation of Intraprocedural and Follow Up Parameters for Mitral Regurgitation Grading after Percutaneous Edge-to-Edge Repair

**DOI:** 10.3390/jcm11092276

**Published:** 2022-04-19

**Authors:** Eduardo Pozo Osinalde, Alejandra Salinas Gallegos, Ximena Gordillo, Luis Nombela Franco, Pedro Marcos-Alberca, Patricia Mahía, Gabriela Tirado-Conte, José Juan Gómez de Diego, Pilar Jiménez Quevedo, Antonio Fernández-Ortíz, Julián Pérez-Villacastín, Jose Alberto de Agustín Loeches

**Affiliations:** 1Cardiology Department, Instituto Cardiovascular, Hospital Clínico San Carlos, 28040 Madrid, Spain; a.salinasgallegos@gmail.com (A.S.G.); ximenagordillo@gmail.com (X.G.); luisnombela@yahoo.com (L.N.F.); pedro.marcosalberca@outlook.es (P.M.-A.); patmahia@gmail.com (P.M.); gabrielatirado@gmail.com (G.T.-C.); josejg@gmail.com (J.J.G.d.D.); patropjq@gmail.com (P.J.Q.); antonio.fernandezortiz@salud.madrid.org (A.F.-O.); jvillacastin@secardiologia.es (J.P.-V.); albertutor@hotmail.com (J.A.d.A.L.); 2Fundación Interhospitalaria para la Investigación Cardiovascular (FIC), Calle Profesor Martín Lagos s/n, 28040 Madrid, Spain; 3Internal Medicine Department, Universidad de La Frontera, Temuco 4781218, Chile; 4Cardiology Department, Hospital Dr. Hernán Henríquez Aravena, Temuco 4781151, Chile; 5Noninvasive Cardiology Department, Instituto Nacional Cardiovascular (INCOR), Lima 15072, Peru

**Keywords:** mitral regurgitation, percutaneous edge-to-edge mitral repair, echocardiography, mitral regurgitation grading, follow up

## Abstract

Background: There is no consensus on the best intraprocedural parameter to evaluate residual mitral regurgitation (MR) after transcatheter edge-to-edge mitral repair (TEER). Thus, our aim was to evaluate the predictive value of different MR parameters from intraprocedural transesophageal echocardiogram (TEE) for grading in consecutive transthoracic echocardiogram (TTE) during the follow up. Methods: All the consecutive patients who underwent TEER with MitraClip between 2010 and 2020 in our center were considered. TEE-derived immediate postprocedural MR parameters were reassessed to blindly compare them with follow up MR grading in sequential TTE. Results: We finally included 88 patients (64.8% males; 76 ± 10 years-old). Significant MR was detected in 14.3% of the cases at 6 months, in similar proportion than at postprocedural at 1 month. Among all the intraprocedural TEE quantitative parameters only additive and maximum VC were associated with significant MR persistence. Moreover, on ROC analysis maximum VC demonstrated an excellent discriminatory power (AUC 0.96; *p* < 0.001) to identify MR ≥ III at 6 months. Thus, a cut-off point of 0.45 cm demonstrated 88% sensitivity and 89% specificity. Conclusion: Among intraprocedural TEE parameters to evaluate residual MR in TEER, maximum and additive VC were the most reliable to predict persistence of significant insufficiency.

## 1. Introduction

Mitral regurgitation (MR) is the second-most-prevalent valvular heart disease in developed countries [1]. Although severe MR is independently associated with mortality in spite of medical therapy [2], a high percentage of patients do not undergo surgical mitral valve (MV) repair or replacement due to high risk [3]. In this clinical scenario, transcatheter edge-to-edge repair (TEER) with the MitraClip^®^ device (Abbott Vascular, Santa Clara, CA, US) has merged as an alternative approach with proven benefit in patients with primary [4] and secondary [5,6] MR.

Clinical impact of this technique has been directly related with the reduction in MR grade. However, as the result of at least a double orifice mitral valve after the device implantation, residual MR is frequently composed by multiple jets, which significantly hampers echocardiographic grading. Actually, complex postprocedural mitral valve anatomy precludes the use of the parameters recommended for noninvasive evaluation of native valve regurgitation [7]. A more recent consensus for the evaluation of valvular insufficiency after percutaneous treatment [8] advocates for a multiparametric approach with integration of many semiquantitative variables: number and direction of the jets, maximum vena contracta (VC) and proximal isovelocity surface area (PISA), etc. Some studies [9,10] have proposed a 3D vena contracta area (VCA) as a precise tool to identify significant residual MR and have argued that it has prognostic impact. However, this technique shows the same limitation in a scenario of multiple residual jets. More recent papers [11,12] advocate the use of pulmonary vein (PV) flow indexes for residual MR assessment due to their impact in postprocedural rehospitalization and mortality. Nevertheless, in the presence of eccentric jets, PV flow may not have a good correlation with residual MR. In light of these contradictory results, this question seems to remain still open.

Therefore, the aim of our study is to evaluate the predictive value of different MR parameters from intraprocedural transesophageal echocardiography (TEE) with grading in consecutive transthoracic echocardiography (TTE) during the follow up in patients that undergo MitraClip implantation.

## 2. Materials and Methods

### 2.1. Patient Population

We retrospectively collected consecutive patients who underwent TEER of the mitral valve with the MitraClip^®^ system between 2010 and 2020 in our tertiary university hospital, with a structured TEER program since late 2016 that resulted in the performance of 20 cases/year. Elegibiliy for this therapy was made by the Heart Team based on the presence of symptomatic moderate-to-severe or severe MR (2D ERO and regurgitant volume > 0.4/0.3 cm^2^ and >60/45 mL, respectively, for primary/secondary etiology) and contraindication for cardiac surgery attending to high STS [13] and EuroScore II [14] risk scores or severe comorbidities. Moreover, all the patients underwent a previous comprehensive mitral study with TEE to evaluate the anatomic suitability for TEER [15]. Therefore, unsuitable valve morphology as well as active endocarditis and intracardiac thrombus were considered major contraindications for the procedure. Written informed consent was obtained from all the patients before the intervention.

The procedure was performed under general anesthesia using TEE and fluoroscopic guidance. After atrial transeptal puncture and device alignment, both MV leaflets were grasped with the clip that is closed to approximate them (Figure 1). Device selection was based on available generation (first 20 cases were Classic, followed by 28 NT interaction and the last 39 third generation devices). The procedure was considered successful if at least a 2-grade reduction in MR was achieved with no significant mitral stenosis. If the result was not satisfactory the device may be repositioned or more clips placed.

Clinical history as well as treatment of the patients were obtained from their medical records. Development of major cardiovascular events, among other variables, were also evaluated reviewing medical records. This study followed the Declaration of Helsinki for human research.

### 2.2. Echocardiographic Evaluation

Intraprocedural echocardiographic guidance was performed by two imaging specialists with large experience in structural heart disease following a standardized protocol [16]. Following current recommendations [8], whenever possible all the suggested parameters were obtained to evaluate MR grading [17] immediately after device deployment: number of residual jets, additive (sum of different jets) and maximum VC, maximum and additive 3D VCA and PV flow pattern (Figure 2). The latter was taken in both right and left superior veins before guide catheter removal from left atrium. Ratios of maximum velocity (Vmax) and velocity time integral (VTI) between systolic and diastolic waves were calculated considering the worse values, and compared with preprocedural values. Echocardiographer assisting the procedure established the residual MR grade based on an integrative evaluation of the aforementioned parameters. For this study, all the parameters were blindly measured again.

All the patients underwent a close follow up with serial clinical and echocardiographic evaluation. More specifically, TTE was performed at 1 and 6 months after the procedure with a comprehensive MR grading by independent imaging specialists.

### 2.3. Statistical Analysis

Continuous variables were expressed as mean ± standard deviation (SD) or median (interquartile range) based on the normality of their distribution, whereas total number (percentages) were used for categorical variables. Population was divided according to the presence of significant residual MR (≥grade III) in the TTE follow up. Comparisons between these two groups were performed with χ^2^ Fisher exact, unpaired Student’s t-test and Mann–Whitney U test as appropriate.

Correlations were evaluated using Pearson rho (r) coefficient. The concordance between diagnosis of significant residual MR during the procedure and at 1 and 6 months was evaluated using the quadratic weighted coefficient Kappa (K) with 95% confidence interval (CI). Agreement between modalities was considered excellent if K > 0.75, good if K was between 0.4 and 0.75, and poor if <0.4 [18]. The concordance of intraprocedural quantitative MR parameters with follow up values was analyzed using intraclass correlation coefficient (ICC) with 95% confidence intervals (CI). A value of ICC < 0.4 was considered poor, 0.4 to 0.75 acceptable, 0.75 to 0.90 good, and >0.90 excellent [19]. ROC curves were constructed to establish the additive and maximum VC cut-off point needed to detect significant MR in the follow up.

A 2-tailed *p* value < 0.05 was considered statistically significant. Analyses was performed with SPSS (version 23.0, SPSS Inc., Chicago, IL, USA) software for Windows.

## 3. Results

### 3.1. Population Characteristics and Procedural Results

A final population of 88 patients (64.8% males) with an age of 76 ± 10 years-old were finally included in the present study (Table 1). The prevalence of cardiovascular risk factors was high, with 72.7% of hypertension, up to 56.8% of the patients had history of ischemic heart disease (19.3% CABG) and 60.2% was in atrial fibrillation. Comorbidity presence was high, particularly chronic renal failure (58.1%) resulting in excessive surgical risks. MR etiology was almost equally distributed between primary (35%) and secondary (44%) types, with less-frequently mixed forms (21%). Regarding baseline echocardiographic evaluation (Table 2), patients showed mild left ventricular systolic dysfunction (LVEF 44.5 ± 15.3%) and dilatation (LVEDVi 71.8 [51.5–102.8] mL/m^2^). Pulmonary artery systolic pressure was significantly elevated (47 ± 18.1 mm Hg) with preserved right ventricular function. Regarding coexistent valvular heart diseases, significant (moderate or severe) tricuspid regurgitation was detected in up to 54.6% of the patients whereas moderate aortic regurgitation or stenosis in only 14% and 3.4% of the cases, respectively. All the patients presented with moderate-to-severe (14.8%) or severe (85.2%) mitral regurgitation but no stenosis (3D valve area 5.3 ± 1.4 cm^2^). MV anatomy was considered optimal in 50% and acceptable in 45.5% of the cases for TEER.

The procedure was considered successful, with at least a 2-grades reduction in MR in 87.5%, and 90.1% had postprocedural MR ≤ II using 1 [1–1.8] clips per case. Residual MR assessed by intraprocedural TEE was classified as: none (3.4%), mild (43.2%), moderate (44.3%), moderate-to-severe (5.7%) and severe (3.4%). No significant stenosis (residual transmitral mean gradient = 2.2 [2–4] mm Hg and 3D mitral valve area = 2.9 ± 0.8 cm^2^) was caused by this therapy. There was no major intraprocedural complications and only one case required early mitral valve replacement due to a posterior leaflet tear. During a median follow up of 16 [6–28] months, 36.5% required hospitalization due to cardiovascular causes and 2.3% of patients died.

### 3.2. Intraprocedural and Follow Up Residual Mitral Regurgitation Assessment

At 6-months follow up residual MR TTE classification was: none (3.6%), mild (37.5%), moderate (44.6%), moderate-to-severe (5.4%) and severe (8.9%). There was an excellent concordance in significant MR detection with intraprocedural (K = 0.74; *p* < 0.001) and 1-month studies (K = 0.923; *p* < 0.001). The persistence of significant MR was associated with anatomic suitability for the procedure (*p* = 0.002) and procedural success (*p* < 0.001) but not with baseline MR grade (*p* = 0.248) or etiology (*p* = 0.183).

Among the different TEE intraprocedural parameters to quantify residual MR only additive and maximum VC were consistently associated with significant MR at both 1- and 6-months follow up TTE (Table 3). Moreover, there was an acceptable concordance for additive (ICC 0.59; *p* < 0.001) as well as maximum (ICC 0.54; *p* = 0.001) VC and significant correlation (ρ 0.45 *p* < 0.001 and ρ 0.38 *p* = 0.001, respectively) between intraprocedural TEE and follow up TTE measurements. When ROC curves were obtained to determine the ability of this intraprocedural values to identify significant MR at 6 months either additive (AUC 0.88, 95%CI 0.76–0.99; *p* = 0.001) and maximum (AUC 0.96, 95%CI 0.95–0.1; *p* < 0.001) VC (Figure 3) showed and excellent performance. As the latter was slightly superior, we selected a maximum VC cut-off point of 0.45 cm that demonstrated 88% sensitivity and 89% specificity to predict persistence of MR ≥ III.

## 4. Discussion

The main findings of our study about the utility of the different intraprocedural TEE quantitative parameters of mitral insufficiency after TEER were as follows. Significant residual MR prevalence remains stably low throughout the mid-term echocardiographic follow-up, reinforcing the long-lasting improvement of this therapy. More importantly, maximum as well as additive VC appeared to be the only isolated parameters significantly associated with persistence of MR ≥ III during the evolution. Interestingly, the cut-off point proposed for maximum VC (0.45 cm) shows that at least one of the jets has VC in the upper limit of the moderate range. Thus, these parameters may be used as single intraprocedural markers of significant residual MR.

As aforementioned, ongoing guidelines recommend an integrative multiparametric approach [8] to determine mitral insufficiency reduction after MitraClip. However, few studies have explored the value of the different quantitative parameters. Dietl et al. [10] firstly described the role of 3D VCA in this clinical scenario. In this retrospective study only 28% of the patients could be included due to lack of availability of adequate 3D color full volume sets in the rest of the cases. Although a significant reduction of 3D VCA was noted after the treatment, this variable was weakly correlated with postprocedural MR grading and only showed an association with a slight improvement of functional class. Avenatti et al. [9] used the same parameter for quantification of residual MR after MitraClip. They found a significant reduction of 3D VCA after the treatment and described an association with postprocedural MR grading and NYHA functional class. Although the authors claimed a good interobserver variability, only 66% of the studies reached enough quality for diagnosis. Therefore, even though this parameter shows promising results, the need for high technical requirements may reduce its use. Moreover, PV flow analysis has promoted awareness to evaluate the TEER result. The first study [11] demonstrated that changes of systolic and systolic/diastolic maximum velocity ratio after the procedure were independently associated with all-cause mortality and hospitalization, whereas improvement in MR grading showed contradictory results. However, they did not evaluate correlation between PV flow and residual MR. A second larger study [11] consistently confirmed that a low postprocedural systolic/diastolic VTI ratio was an independent predictor of major adverse cardiovascular events during the follow up. Similar results were found for systolic/diastolic maximum velocity ratio. Although persistence of significant MR was associated with these variables it was not included in multivariate analysis. Anyhow, PV flow patterns are also dependent on many aspects such as selection of the pulmonary vein for analysis (especially in eccentric jets) and changes in left atrial pressure, that may affect their reproducibility. In this regard, chronicity of the MR with sometimes very large LA volumes, might make assessment difficult, as the typical systolic flow reversal described for severe MR might not be present. In summary, these new quantitative parameters are far from be free of limitations and they are not available in all of the cases. Additionally, they were also evaluated in our study, and did not find significant correlation with follow up MR grading, in probable relation with the previously mentioned limitations. Thus, a widely accessible and simple parameter such as VC may become a useful approach to immediate result estimation after MitraClip deployment.

### Limitations

The present study is not free from limitations. Firstly, the retrospective evaluation precludes the systematic use of all of the available quantitative parameters. In this regard, 3D VCA and PV flow pattern could not be obtained in some patients (59% and 38%, respectively) due to technical limitations, as occurred in previous studies. PISA has not been considered for analysis either due to scarce availability among our patients or relevant limitations for its measurement in this clinical scenario. Moreover, during the procedure we performed a comprehensive TEE evaluation but parameter analysis could only be performed based on available projections. Further derived from the study design, a collection of past medical history and events from electronic medical records ensured that there were no missing complications during the evolution. In any case, prognostic impact of the different parameters was beyond the scope of the present work. Besides, although MR grading of the follow up TTE has been previously performed by an independent expert on cardiac imaging, a bias derived from postprocedural evaluation cannot be totally excluded. In this regard, the different quantitative parameters were blindly re-measured to minimize influence in subsequent MR grading.

## 5. Conclusions

When intraprocedural quantitative parameters to assess residual MR after TEER were individually evaluated, only maximum and additive VC were associated with persistence of significant insufficiency. As the former variable showed the best performance, measurement of maximum VC may constitute a good single estimation of significant MR right after device deployment.

## Figures and Tables

**Figure 1 jcm-11-02276-f001:**
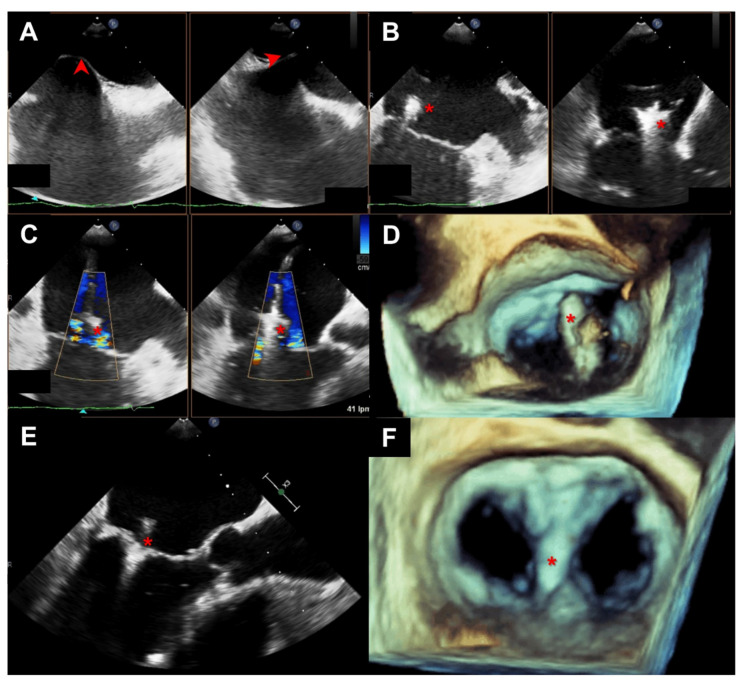
Intraprocedural TEE guidance during TEER with MitraClip. Firstly, transeptal puncture at the superior and posterior portion of fosa ovalis (*arrow head*) should be monitored (**A**). Afterwards MitraClip (*asterisk*) delivery catheter is deflected towards the mitral valve (**B**) and directed against the origin of the regurgitant jet (**C**). Three-dimensional view of the mitral valve helps on perpendicular orientation of the device (**D**). After grasping of both mitral leaflets (**E**) a double-orifice valve results from the procedure (**F**). TEE: transesophageal echocardiogram; TEER: transcatheter edge-to-edge mitral repair.

**Figure 2 jcm-11-02276-f002:**
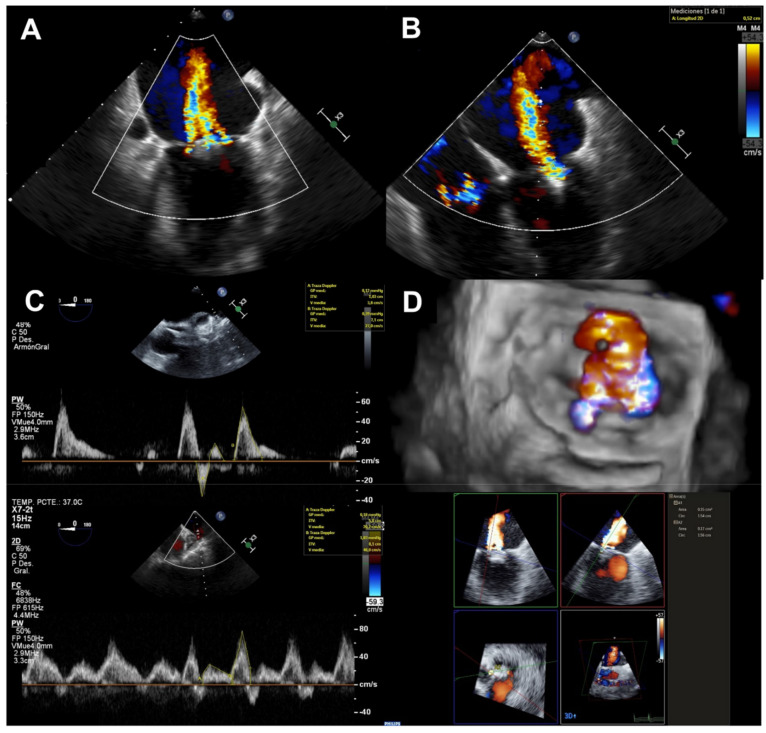
Multiparametric TEE evaluation of residual MR after MitraClip deployment. Significant residual MR assessment with the different quantitative parameters: number of jets (**A**), maximum VC (**B**), changes in PV flow (**C**) and 3D VCA of the different jets (**D**). TEE: transesophageal echocardiogram; MR: mitral regurgitation; VC: vena contracta; PV: pulmonary vein; VCA: vena contracta area.

**Figure 3 jcm-11-02276-f003:**
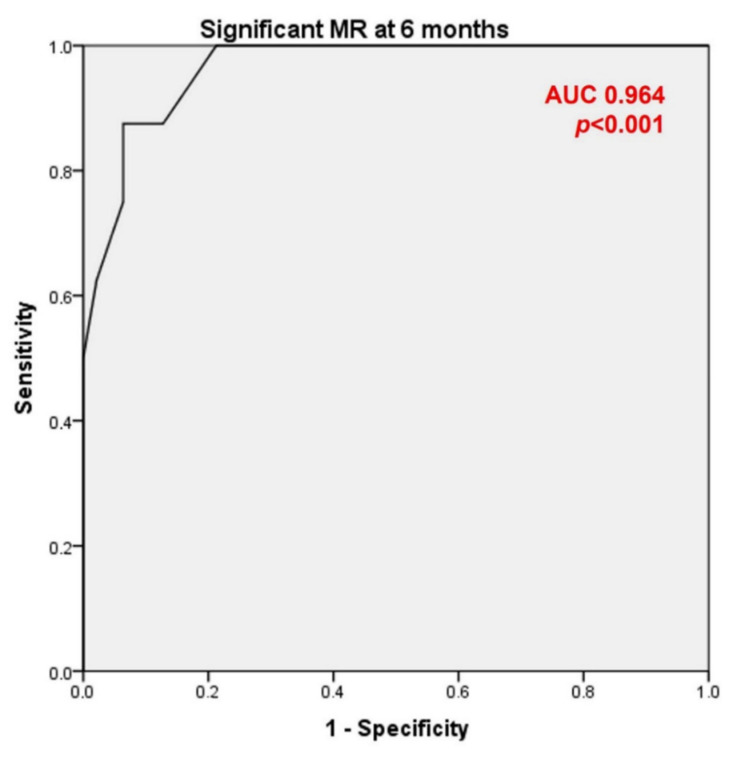
ROC curve of maximum VC for prediction of significant MR at 6 months. ROC: receiver operating curve; VC: vena contracta; MR: mitral regurgitation; AUC: area under the curve.

**Table 1 jcm-11-02276-t001:** Baseline clinical characteristics.

Age (years)	76.2 ± 10
Male (%)	57 (64.8)
Cardiovascular risk factors (%):	
-Hypertension	64/88 (72.7)
-Diabetes	30/88 (34)
-Dyslipidemia	49/88 (55.7)
-Obesity	17/(8 (21)
Cardiovascular history:	
-Ischemic heart disease	50/88 (56.8)
-Previous CABG	17/88 (19.3)
-AF	53/88 (60.2)
Comorbidities:	
-COPD (%)	20/88 (22.7)
-Chronic renal failure (%)	50/88 (58.1)
-Cancer (%)	8/88 (9.9)
-Stroke (%)	7/88 (8)
-Peripheral vascular disease (%)	15/88 (17)
MR etiology (%)	
-Primary	31/88 (35)
-Secondary	38/88 (44)
-Mixed	18/88 (21)
NYHA III-IV (%)	78/88 (88.6)
Surgical risk scores:	
-EuroScore II	9.14 ± 8.7
-STS score	6.4 ± 5.4

CABG: coronary artery bypass grafting; AF: atrial fibrillation; COPD: chronic obstructive pulmonary disease; MR: mitral regurgitation, STS: society of thoracic surgeons.

**Table 2 jcm-11-02276-t002:** Baseline echocardiographic characteristics.

LVEF (%)	44.5 ± 15.3
LVEDD (mm)	56.9 ± 10.4
LVESD (mm)	42.4 ± 13.6
LVEDVi (mL/m^2^)	71.8 [51.5–102.8]
LVESVi (mL/m^2^)	38 [21.1–72.8]
LAVi (mL/m^2^)	53 [45.8–66]
TAPSE (mm)	18.7 ± 4.1
FAC (%)	38.7 ± 9.7
PASP (mm Hg)	47 ± 18.1
3D MVA (cm^2^)	5.3 ± 1.4
Transmitral mean gradient (mm Hg)	2 [1–2.2]
Mitral pressure half time (ms)	94 ± 34
MR grade (%):	
-Moderate-to-severe (III/IV).	13 (14.8)
-Severe (IV/IV)	75 (85.2)
Anatomic suitability for TEER (%)	
-Optimal	44 (50)
-Acceptable	40 (45.5)
-Unfavorable	4 (4.5)

LVEF: left ventricular ejection fraction; LVEDD: left ventricular end-diastolic diameter; LVESD: left ventricular end-systolic diameter; LVEDVi: left ventricular end-diastolic volume indexed; LVESVi: left ventricular end-systolic volume indexed; LAVi: left atrial volume indexed; TAPSE: tricuspid annular plane systolic excursion; FAC: fractional area change; PASP: pulmonary artery systolic pressure; MVA: mitral valve area; MR: mitral regurgitation; TEER: transcatheter edge-to-edge mitral repair.

**Table 3 jcm-11-02276-t003:** Differences in intraprocedural TEE quantitative parameters in relation with significant MR in follow-up TTE.

	1 Month			6 Months		
	Significant MR (n = 7; 8%)	Non-Significant MR (n = 81; 92%)	*p*	Significant MR (n = 8; 9%)	Non-Significant MR (n = 80; 91%)	*p*
N° Jets	2 [2–3]	2 [1–3]	0.966	2 [2–2.75]	2 [1–3]	0.654
VC add (cm)	0.79 [0.48–1]	0.44 [0.31–0.65]	0.022	0.9 [0.6–1.13]	0.4 [0.3–0.59]	<0.001
VC max (cm)	0.47 [0.35–0.54]	0.32 [0.25–0.4]	0.004	0.52 [0.47–0.6]	0.3 [0.22–0.4]	<0.001
3D VCA add (cm^2^)	0.23 [0.12–0.42]	0.18 [0.15–0.29]	0.754	0.42 [0.25–0.42]	0.18 [0.13–0.29]	0.041
3D VCA max (cm^2^)	0.15 [0.12–0.23]	0.16 [0.13–0.19]	0.656	0.21 [0.15–0.21]	0.17 [0.13–0.19]	0.192
PV syst Vmax post (cm/s)	37.4 [28.7–54.8]	35.6 [28.8–51.1]	0.985	51.5 [30.4–66.6]	36.8 [30.2–51.1]	0.343
PV syst VTI post (cm)	8.2 [5.2–11.2]	8.1 [5.9–11.5]	0.799	9.5 [4.9–14.4]	8.1 [5.9–10.9]	0.9
PV syst/diast Vmax	1.1 [0.69–1.22]	0.97 [0.69–1.34]	0.757	0.9 [0.66–1.19]	1 [0.75–1.43]	0.377
PV syst/diast VTI	0.73 [0.39–1.76]	0.82 [0.61–1.41]	0.513	0.42 [0.35–0.78]	0.9 [0.63–1.36]	0.029
Δ PV syst/diast Vmax	1.06 [0.44–1.79]	0.98 [0.45–1.61]	0.712	0.84 [0.5–1.98]	0.9 [0.47–1.28]	0.834
Δ PV syst/diast VTI	0.81 [0.03–1.53]	1.04 [0.42–1.5]	0.515	0.42 [−0.04–1.6]	0.66 [0.24–1.37]	0.444

TEE: transesophageal echocardiogram; TTE: transthoracic echocardiogram; VC: vena contracta; add: additive; max: maximum; VCA: vena contracta area; PV: pulmonary vein; VTI: velocity–time integral; syst: systolic; Vmax: maximum velocity; post: postprocedural; diast: diastolic; Δ: difference.

## Data Availability

Data supporting our results is kept in an anonymized database.

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
