# Peer review of "Correlation of Intraprocedural and Follow Up Parameters for Mitral Regurgitation Grading after Percutaneous Edge-to-Edge Repair"

_jcm, 2022, doi:10.3390/jcm11092276_

Round 1
Reviewer 1 Report
The submitted manuscript with title: "Correlation of intraprocedural and follow up parameters for MR grading after percutaneous edge-to-edge repair" , despite its retrospective nature and the small number of patients analyzed, achieved to demonstrate an additional significance of VC as an intraprocedural TEE parameters for further evaluation of MR in TEER.
The statistical and methods part are clearly presented.
However, there are some open points . which I would love to emphasise:
a) What kind of Mitraclip-Device was used in your patient collective? I could find any infos about that. --> Please do specify it and the anatomical selection for it .
b) Despite the TEE parameters at Table 2, did you also define Post-clip MVOA, post-clip TMPG ? As a combination of these parameters with VC would have strengthened the discriminatory power!
c)At the baseline characteristics missing description of TR-presence. How was it in your study? As it has an important significance for all-cause mortality after TEER.
d)Is there any pre-procedural TEE measuring of the tethering angle of posterior leaflet ? If yes, do define it. It presents an important significance for device failure.
Author Response
The submitted manuscript with title: "Correlation of intraprocedural and follow up parameters for MR grading after percutaneous edge-to-edge repair" , despite its retrospective nature and the small number of patients analyzed, achieved to demonstrate an additional significance of VC as an intraprocedural TEE parameters for further evaluation of MR in TEER.
The statistical and methods part are clearly presented.
We really appreciate your deep review that will certainly improve the quality of the paper and enhance its clinical relevance.
However, there are some open points . which I would love to emphasise:
- a) What kind of Mitraclip-Device was used in your patient collective? I could find any infos about that. --> Please do specify it and the anatomical selection for it .
In the first 20 cases classic device was used, afterwards Mitraclip NT was implanted in the next 28 patients and the last 39 received the third generation devices (26 only XTr, 11 only NTr and 2 patients both XTr and NTr). In the latter cases device selection was based on leaflet length and basal mitral valve area. I brieft description of this technical aspects has been added to “Patient population” paragraph (lines 83-85)
- b) Despite the TEE parameters at Table 2, did you also define Post-clip MVOA, post-clip TMPG ? As a combination of these parameters with VC would have strengthened the discriminatory power!
We do collected tridimensional vena contracta area (3D VCA) but as it is explained in Table 3 there were no statistical differences in significant MR neither at 1 month nor at 6 months echocardiographic follow up. Regarding mitral stenosis markers, we do not included them as predictors of significant MR at follow up. We report absence of significant stenosis with a residual residual transmitral mean gradient of 2.2 [2-4] mm Hg (line 159). We have included the post-procedural mitral valve area by 3D planimetry to reinforce this point (line 159).
c)At the baseline characteristics missing description of TR-presence. How was it in your study? As it has an important significance for all-cause mortality after TEER.
Indeed, significant TR is a well-known determinant of prognosis. We have report its prevalence among with the other valvular heart diseases in the text (lines 152-154).
d)Is there any pre-procedural TEE measuring of the tethering angle of posterior leaflet ? If yes, do define it. It presents an important significance for device failure.
Unfortunately this measurement is not available in our patients. At the beginning of the program we used to consider tenting and tethering of the mitral leaflets but as long as we have gained experience we do not take in account these parameters for patient selection.
Reviewer 2 Report
I've read with interest the manuscript "Correlation of intraprocedural and follow up parameters for mitral regurgitation grading after percutaneous edge-to-edge repair" by Osinalde and coworkers.
My observations are the following:
- typo: "merged" instead of "emerged" (Line 45)
- typo: "an scenario" instead of "a scenario". Also please check "limitation" (Line 59)
- typo: "At the light of this contradictory results"instead of "In light of these contradictory results" (Lines 62, 63)
- typo: "the aim of our study is "instead of "the aim of this study was" (Line 64)
- Indication for TEER was "presence of symptomatic moderate-to-severe or severe MR and contraindication for cardiac surgery" (Lines 72, 73). You must clarify your inclusion criteria. Moderate mitral MR is not an indication for surgery in the current AHA or ESC guidelines. What were actually your inclusion criteria? You don't specify any quantitative parameters. Your resulting patient cohort may have been comprised of less ill patients (with moderate MR) and those with severe MR and contraindication for surgery, which would not result in a homogenous patient population.
- ortography: "summatory"(Line 104). Consider replacing with "additive".
- "Interventional echocardiographer established" consider revising
- What were the reasons for the surgical contraindications? Consider adding a table to detail this. The mean Euroscore and STS scores in Table 1 are not high-enough to justify contraindicating those patients for surgery. Those patients had >90% chances of survival after conventional surgical MVR
- Please replace "Ictus" in Table 1 with "Stroke"
- Please specificy what criteria did you use to grade intraprocedural MR (Lines 156-158). More generally, what quantitative or qualitative parameters did you use to classify MR in double-orifice patients?
- It seems to me that the whole question about the correlation of intraprocedural ultrasound parameters with late follow-up grading is a bit unclear: you measure intraprocedural ultrasound parameters, because you say it's unclear which one is the best for grading MR. Then, at follow-up, you classify residual MR (on the basis of which parameters? VC/PISA, qualitatively?). Then you draw correlations between intraprocedural parameters and residual MR grading. It's totally unclear how you grade residual MR in double-orifice TEER patients and, basically, this is crucial for your study conclusions.
Author Response
I've read with interest the manuscript "Correlation of intraprocedural and follow up parameters for mitral regurgitation grading after percutaneous edge-to-edge repair" by Osinalde and coworkers.
We thank you for the careful reading and meticulous review of our article. Hereafter, we will answer your comments and requests in detail trying to increase the understanding of the findings reported in the paper.
My observations are the following:
typo: "merged" instead of "emerged" (Line 45)
typo: "an scenario" instead of "a scenario". Also please check "limitation" (Line 59)
typo: "At the light of this contradictory results"instead of "In light of these contradictory results" (Lines 62, 63)
typo: "the aim of our study is "instead of "the aim of this study was" (Line 64)
All the aforementioned typos have been corrected.
Indication for TEER was "presence of symptomatic moderate-to-severe or severe MR and contraindication for cardiac surgery" (Lines 72, 73). You must clarify your inclusion criteria. Moderate mitral MR is not an indication for surgery in the current AHA or ESC guidelines. What were actually your inclusion criteria? You don't specify any quantitative parameters. Your resulting patient cohort may have been comprised of less ill patients (with moderate MR) and those with severe MR and contraindication for surgery, which would not result in a homogenous patient population.
We strictly comply with the ongoing guidelines for the management of our patients so we do not indicate any procedure unless is considered beneficial. Inclusion criteria are explained under “Patient population” paragraph. The reasons why moderate-to-severe (grade 3+) mitral regurgitation was included in the study are two. First, there is disparity between AHA and ESC guidelines in ERO and regurgitant volume cut-off points in secondary mitral regurgitation. We follow European recommendations with lower values, which are not considered severe in the American recommendations. We have specified the quantitative parameters considered for the inclusion (lines 73-74) Second, pivotal studies of TEER (EVEREST, MITRA-FR and COAPT) included patients with grade 3 and 4 mitral regurgitation.
ortography: "summatory"(Line 104). Consider replacing with "additive".
As requested, we have changed all the summatory terms by additive.
"Interventional echocardiographer established" consider revising
This statement was rephrased.
What were the reasons for the surgical contraindications? Consider adding a table to detail this. The mean Euroscore and STS scores in Table 1 are not high-enough to justify contraindicating those patients for surgery. Those patients had >90% chances of survival after conventional surgical MVR
I respectfully disagree with this comment. 2021 ESC guidelines for the management of valvular heart disease consider that patients with an EuroScore II >8% are at high risk for cardiac surgery and our population was above this cut-off point. Moreover, our patients were elderly (76.2±10 years-old) and with a high proportion of previous CABG (19.3%) and comorbidities (COPD 22.7%, chronic renal disease 58.1%...). In any case, every patient that was considered for percutaneous mitral valve repair was evaluated by our Heart Team (composed by clinical cardiologists, interventional cardiologists, specialists in cardiac imaging and cardiac surgeons) who formally rejected them for cardiac surgery. This statement was mentioned under “Patient population” (lines 70-74).
Please replace "Ictus" in Table 1 with "Stroke"
This change was done.
Please specify what criteria did you use to grade intraprocedural MR (Lines 156-158). More generally, what quantitative or qualitative parameters did you use to classify MR in double-orifice patients?
As mentioned under “Introduction” (lines 51-56) there are recent recommendations for the post-procedural grading of MR that advocate for an integrative evaluation of the multiple quantitative and qualitative parameters. In this way, echocardiographer assisting the procedure considered number of residual jets, additive and maximum VC, maximum and additive 3D VCA and PV flow pattern (lines 101-106). Actually, the main aim of our study is to establish which one of the aforementioned criteria predict better residual MR in the follow up.
It seems to me that the whole question about the correlation of intraprocedural ultrasound parameters with late follow-up grading is a bit unclear: you measure intraprocedural ultrasound parameters, because you say it's unclear which one is the best for grading MR. Then, at follow-up, you classify residual MR (on the basis of which parameters? VC/PISA, qualitatively?). Then you draw correlations between intraprocedural parameters and residual MR grading. It's totally unclear how you grade residual MR in double-orifice TEER patients and, basically, this is crucial for your study conclusions.
This is the key point of the study. MR grading is challenging immediately after MitraClip implantation due to several reasons. First, a double-orifice (at least) valve hampers the use of classical parameters for assessment in native valves. Moreover, changes in loading conditions may also affect the evaluation. In such scenario recent guidelines recommend an integrative multiparametric evaluation. However, sometimes these parameters are discordant, so our aim is to select the single parameter more precise to define MR grade after implantation. For this purpose, as explained on paragraph “Echocardiographic evaluation” (lines 101-115) of “Material and Methods”, a careful blind reassessment of the different intraprocedural parameters was correlated with MR grading at follow up previously done by an independent imaging specialist.
Reviewer 3 Report
I read with interest the study "Correlation of intraprocedural and follow up parameters for 2 mitral regurgitation grading after percutaneous edge-to-edge 3 repair" by Eduardo Pozo Osinalde et. al.
With their work, the authors find a correlation of multiple parameters of MR assessment during TEER with long term MR results. Furthermore, they claim to have identified V.c. measurements as best independent predictors for the result at 1 and at 6 months.
The work is of general interest, as TEER has become widely used and its use is still expanding, while data on good intra-procedural predictors for favourable results are scarce due to few studies with low patient numbers.
In this regard, the study at hand offers some new insights, but it suffers greatly from (A) also low numbers and (B) from a somewhat vague structure. Still, I think there is merit in this work, if the issues raised below can be adequately addressed.
Major issues:
- the population size of n=88 patients seems quite low for a time frame of 10 years.
- Furthermore, and with the former issue, it is unclear if n=88 really is the whole population, or the population that was available for analysis, as some dropped out due to loss-of-FUP. Please indicate. With that said, a FUP of 100 % seems quite surprising. If the n=88 represents the population available for analysis at FUP, please also indicate the whole population demographics in Table 1
- Please also compare baseline characteristics within the two subgroups and show the data in table 1.
- Please indicate the difference between summatory and/or maximum VC. Was biplane V.c. used?
- Also in line 89 (see below on "typo"): I suggest to not advocate the "surgical view" as standard. I think better would be "A Three-dimensional" only. The surgeon's view may be used at some centers, but the more logical approach is the cardiologist's view, as this is the 3D image that is generated when oeprating from the inter-comissural view as main plane.
- Figure 1: Figure 1A "IAS puncture" would be even better understandable if it showed a view with the aorta, or even better the aortic valve (this is anterior) as point of reference and the tenting being away from there, i.e. posterior.
- Figure 1: The clips seem not "aligned" to the Echo view ("foreshortened"). So, using Philips echo machines, either the primary angle is off, or, the clip's orientation is.
- I strongly suggest to NOT use citation [16] as reference for a guiding protocol. This is outdated. Today's MitraClip is guided differently. I suggest to check, if "EuroIntervention 2014;10:884-886. DOI: 10.4244/EIJV10I7A150" would also fits the local practice. The publication "doi:10.1016/j.mayocp.2018.10.007" is a little new, but relies (maybe too) heavy on 3d volume guidance.
- With n=88 patients in 10 years (s.a.), one might challenge the "large experience" of 2 imaging specialists, as some consider that TEER should only be available in centers with one operator having experience of n=20 or more cases p.a. (here: experience=80/10/2= 4 cases p.a.). I state this also to point out the importance of the point made above: really n=88 with 100 % FUP?
- Later, we learn, that 3D VCA was not available in all cases. Please indicate the exact numbers.
- Please indicate more clearly, how the primary endpopint is defined: is it MR 3+ and MR 4+, or MR 4+, only?
- While this is not the focus of this study, please do a paired analysis on outcomes for the whole cohort. This would back up the discussion's statement "Significant residual MR prevalence remains stably low throughout the midterm echocardiographic follow-up, reinforcing the long-lasting improvement of this therapy". Also do this testing on the subgroups (where with the latter no significance would be expected in the subgroup of persisting severe MR).
- In the results section, the MR grades at 6mo FUP are delineated. However, Table 3 also offers 1 month data. Please indicate the results for this assessment, too. Is there a shift in numbers of significant MR?
- Therefore, please indicate the numbers of patients in each subgroup in table 3
- The statement in the discussion "More importantly maximum as well as summatory VC appeared to be the only isolated parameters significantly associated with persistence of MR ≥III during the evolution. Thus, these parameters may be used as single intraprocedural markers of significant residual MR." is a little bit difficult at the moment: first, referencing to above mentioned remarks, it is not clear how V.c. was devised exactly. Furthermore, the PISA method was not used. I personally find this ok, but maybe not everybody thinks this way. In any case, if possible, an analysis on PISA would be greatly appreciated by the community, since its use seems widely accepted (even after TEER), although there are major limitations to this method. At least, this should be discussed.
- In the discussion, line 233 ff, the sentence "Anyhow, PV flow patterns are also dependent on many aspects such as selection of the pulmonary vein for analysis (especially in eccentric jets) and changes in left atrial pressure, that may affect their reproducibility" is not entirely correct. The changes in LAP are the foundation of its usefulness, not a confounder. Instead, I would point out, that different loading conditions, chronicity of the MR with sometimes very large LA volumes, might make assessment difficult, as the typical systolic flow reversal described for severe MR might not be present, albeit the severity of the disease. Nevertheless, in this study, there is a sisgnificant difference at 6 months.
- Please discuss the cut-off values calculated. Is it reasonable? Is this cut-off an implication on (change of) therapy, i.e. more clips?
Minor issues:
- There is a typo in line 27: "posprocedural"
- Typo in line 89: "Tridimensional..."
- Line 153: How was "optimal" MV anatomy defined? ("MV anatomy was considered optimal in 50% and acceptable in 45.5% of the cases for TEER.")
- Table 3: what does TVI stand for? Furthermore: Did you mean VTI, i.e. velocity-time integral? (VTI is also used in the text)
Author Response
I read with interest the study "Correlation of intraprocedural and follow up parameters for 2 mitral regurgitation grading after percutaneous edge-to-edge 3 repair" by Eduardo Pozo Osinalde et. al.
With their work, the authors find a correlation of multiple parameters of MR assessment during TEER with long term MR results. Furthermore, they claim to have identified V.c. measurements as best independent predictors for the result at 1 and at 6 months.
The work is of general interest, as TEER has become widely used and its use is still expanding, while data on good intra-procedural predictors for favourable results are scarce due to few studies with low patient numbers.
In this regard, the study at hand offers some new insights, but it suffers greatly from (A) also low numbers and (B) from a somewhat vague structure. Still, I think there is merit in this work, if the issues raised below can be adequately addressed.
We really appreciate your effort in paper review that will certainly increase the quality of the exposition of our findings and redefine their clinical implications.
Major issues:
the population size of n=88 patients seems quite low for a time frame of 10 years.
Furthermore, and with the former issue, it is unclear if n=88 really is the whole population, or the population that was available for analysis, as some dropped out due to loss-of-FUP. Please indicate. With that said, a FUP of 100 % seems quite surprising. If the n=88 represents the population available for analysis at FUP, please also indicate the whole population demographics in Table 1
Please also compare baseline characteristics within the two subgroups and show the data in table 1.
All the consecutive patients included in our program in the reported timeframe was 88. Certainly, in the first years the use of this therapy was anecdotal with only 8 cases performed in the 2010-2015 period. Since late 2016 a structured TEER program was established with a mean cases rate of 20 implants per year. We have explained this point under “Patient population” (lines 72-73). We could remove the first cases but we consider that reporting the full experience readers get a more realistic view of the learning curve of the procedure. Consequently, there are no subgroups to compare.
Please indicate the difference between summatory and/or maximum VC. Was biplane V.c. used?
We added the value of the vena contracta of the different jets to get the summatory or additive value while we took the vena contracta of the larger jet to ge the maximum. This has been explained in “Echocardiographic evaluation” (line 107).
Also in line 89 (see below on "typo"): I suggest to not advocate the "surgical view" as standard. I think better would be "A Three-dimensional" only. The surgeon's view may be used at some centers, but the more logical approach is the cardiologist's view, as this is the 3D image that is generated when operating from the inter-comissural view as main plane.
He have removed the term “surgical view” as requested.
Figure 1: Figure 1A "IAS puncture" would be even better understandable if it showed a view with the aorta, or even better the aortic valve (this is anterior) as point of reference and the tenting being away from there, i.e. posterior.
Certainly, it may be helpful to mark aortic valve and superior vena cava position for IAS puncture location; however, we did not do it because a composite image with several markers may result confusing.
Figure 1: The clips seem not "aligned" to the Echo view ("foreshortened"). So, using Philips echo machines, either the primary angle is off, or, the clip's orientation is.
Clip is of the angle in Image B. Nevertheless, his panel try to explain device alineation process that is completed in Panel D, resulting in a good grasping (E and F)
I strongly suggest to NOT use citation [16] as reference for a guiding protocol. This is outdated. Today's MitraClip is guided differently. I suggest to check, if "EuroIntervention 2014;10:884-886. DOI: 10.4244/EIJV10I7A150" would also fits the local practice. The publication "doi:10.1016/j.mayocp.2018.10.007" is a little new, but relies (maybe too) heavy on 3d volume guidance.
We have changed the reference as requested.
With n=88 patients in 10 years (s.a.), one might challenge the "large experience" of 2 imaging specialists, as some consider that TEER should only be available in centers with one operator having experience of n=20 or more cases p.a. (here: experience=80/10/2= 4 cases p.a.). I state this also to point out the importance of the point made above: really n=88 with 100 % FUP?
As it was explained in the answer to the first question, after an initial period of 6 years of sporadic cases in the last 4 years we have performed 80 cases so we accomplished with the requirement of cases per year.
Later, we learn, that 3D VCA was not available in all cases. Please indicate the exact numbers.
3D VCA was only available in 36 cases (41%). In the rest of the cases the parameter was not available because residual MR was absence/mild or Color 3D was not available. This limitation was already commented in “Discussion”. Moreover, we have added in this section (lines 252-253). As also commented in the same section a similar problem was reported in previous studies.
Please indicate more clearly, how the primary endpopint is defined: is it MR 3+ and MR 4+, or MR 4+, only?
The primary endpoint was persistence of significant MR, defined as ≥3+, in the follow up echocardiogram. This was already mentioned in “Statistical analysis” as “Population was divided according to the presence of significant residual MR (≥grade III) in the TTE follow up” (lines 127-128).
While this is not the focus of this study, please do a paired analysis on outcomes for the whole cohort. This would back up the discussion's statement "Significant residual MR prevalence remains stably low throughout the midterm echocardiographic follow-up, reinforcing the long-lasting improvement of this therapy". Also do this testing on the subgroups (where with the latter no significance would be expected in the subgroup of persisting severe MR).
The aim of our study was to compared the different intraprocedural TEE quantitative MR grading parameters with significant mitral insufficiency at follow up TTE. The statement mentioned only refers to the stable reduction on MR. Therefore, prognosis analysis, although with relevant clinical implications, goes beyond the scope of the present paper.
In the results section, the MR grades at 6mo FUP are delineated. However, Table 3 also offers 1 month data. Please indicate the results for this assessment, too. Is there a shift in numbers of significant MR?
Therefore, please indicate the numbers of patients in each subgroup in table 3
As pointed out in the text (“There was an excellent concordance in significant MR detection with intraprocedural (K= 0.74; p< 0.001) and 1-month studies (K= 0.923; p< 0.001)”) prevalence of significant MR was stable between TTE at 1 and 6 months after the procedure. In any case, we have added the values to Table 3, as requested.
The statement in the discussion "More importantly maximum as well as summatory VC appeared to be the only isolated parameters significantly associated with persistence of MR ≥III during the evolution. Thus, these parameters may be used as single intraprocedural markers of significant residual MR." is a little bit difficult at the moment: first, referencing to above mentioned remarks, it is not clear how V.c. was devised exactly. Furthermore, the PISA method was not used. I personally find this ok, but maybe not everybody thinks this way. In any case, if possible, an analysis on PISA would be greatly appreciated by the community, since its use seems widely accepted (even after TEER), although there are major limitations to this method. At least, this should be discussed.
We have explained in a previous answer as well as in the manuscript the way these measurements were taken. Among all the single parameters analyzed VC appeared the most reliable for the detection of significant MR in the follow up. Certainly, PISA was not considered in our analysis. Apart of the scarce availability of this parameter in our series some limitations, such as determination in a multiple jet scenario and difficulties on measurement due to shadowing from the devices, have precluded its evaluation in our study. We have mentioned this issue under “Limitations” (lines 253-255)
In the discussion, line 233 ff, the sentence "Anyhow, PV flow patterns are also dependent on many aspects such as selection of the pulmonary vein for analysis (especially in eccentric jets) and changes in left atrial pressure, that may affect their reproducibility" is not entirely correct. The changes in LAP are the foundation of its usefulness, not a confounder. Instead, I would point out, that different loading conditions, chronicity of the MR with sometimes very large LA volumes, might make assessment difficult, as the typical systolic flow reversal described for severe MR might not be present, albeit the severity of the disease. Nevertheless, in this study, there is a significant difference at 6 months.
With this statement we are not trying to underestimate the value of PV flow patterns in MR evaluation but to put in context that dependance on parameters you mentioned may justify its lower predictive value. To further explained this point we have added your comment to the manuscript (lines 243-245). We have found differences in PV flow regarding persistence of significant mitral insufficiency at 6 months but VC was better for discrimination.
Please discuss the cut-off values calculated. Is it reasonable? Is this cut-off an implication on (change of) therapy, i.e. more clips?
We have discussed the cut-off value in the text (lines 216-218) and proposed an explanation for it. Regarding procedural implications of these parameters, they may change the therapy management but as this study was retrospective we considered this statement highly hypothetical so we did not include it in the Discussion.
Minor issues:
There is a typo in line 27: "posprocedural"
Typo in line 89: "Tridimensional..."
We have corrected the aforementioned typos.
Line 153: How was "optimal" MV anatomy defined? ("MV anatomy was considered optimal in 50% and acceptable in 45.5% of the cases for TEER.")
As mentioned on “Patient population”, all the patients underwent a previous comprehensive anatomic mitral study with TEE for suitability of TEER and classified following Boekstegers et al recommendations [15] in ideal (optimal), to be considered (acceptable) and not recommended.
Table 3: what does TVI stand for? Furthermore: Did you mean VTI, i.e. velocity-time integral? (VTI is also used in the text)
We have corrected this mistake.
Round 2
Reviewer 2 Report
No further comments.
Author Response
Thank you very much for your review.
Reviewer 3 Report
I read the revised manuscript to the study "Correlation of intraprocedural and follow up parameters for 2 mitral regurgitation grading after percutaneous edge-to-edge 3 repair" by Eduardo Pozo Osinalde et. al.
All my comments have been addressed.
Please not, that there is now a problem in table 3, since the variables have been changed:
I think, in line 2, it should say "VC add (cm)" while it is "VC summ (cm)" at the moment. Furthermore, in line 4, there is no need for any additional term like "ass" or "sum" to 3D VCA, due to the nature of the measurement. So it probably should only be "3D VCA (cm2)"
Author Response
Regarding Table 3, we have changed the term "summ" in line 2 by "add". However, we have kept the term "add" in line 4 because we analyzed maximum as well as additive (as the summ of the different jets) 3D VCA.
We really appreciate your comments.